# An Overview of the Dynamics of Relative Research Performance in Central-Eastern Europe Using a Ranking-Based Analysis Derived from SCImago Data

**Ioan Ianoş [1] and Alexandru-Ionuţ Petrişor [2],\*** 

[1]  Interdisciplinary Centre of Advanced Research on Territorial Dynamics, University of Bucharest, 050663 Bucharest, Romania; office@cicadit.ro

[2]  Doctoral School of Urban Planning, Ion Mincu University of Architecture and Urbanism, 010014 Bucharest, Romania

\*  Correspondence: alexandru_petrisor@yahoo.com; Tel.: +40-213-077-191

**Abstract:** In recent times, rankings seem to play an increasingly important role, influencing the lives of individual researchers or academics and their institutions. Individual and institutional rankings used for promotion and research or academic funding seem to illustrate more and more the "publish or perish" mantra, relying sometimes almost exclusively on publications and their citations. Eastern Europe found itself part of this new world after a period of isolation, uneven for the countries within the area. The present study uses SCImago data to perform a regional analysis of individual and aggregated domains, for individual countries and the entire region, based on a novel "adjusted citation index", in order to measure the performance and identify, using correlations with additional data and information, the mechanisms that can increase the research performance of a country. In a nutshell, the results indicate that the national research policies are responsible for performance. Adaptive research policies simulate a real performance, in comparison with more restrictive ones, which are more likely to stimulate unethical behaviors such as self-citations or citation stacking, especially when used for the assessment of researchers. The importance of the findings lies in the possibility of replicating the methodology, adapting it to different spatial scales.

**Keywords:** citation and self-citation effects; adjusted citation index; R&D expenditure; ranking-based analysis; research evaluation; regional ranking changes

## 1. Introduction

During the last two decades there was a strong shift in the assessment of the research performance [1–4] at the different scales, starting from the country level and moving to institutions, journals and researchers. As a consequence, a rich recent literature has developed to propose different methodologies, tools and quantitative and qualitative indicators for ranking the universities and research institutions and establishing criteria for assessing the assessment of researchers [5–12].

At the same time, many authors pinpointed the limits of the assessment criteria that are currently used, especially regarding the excessive use of some indexes for measuring research performance and the promotion of researchers [13–19]. In this context, the San Francisco Declaration on Research Assessment (DORA), signed by 1500 organizations and over 15,000 prominent individuals [20], has played an important role in shifting the current assessment patterns and promoting the best practices in this field. This initiative strongly recommends the need to eliminate the current journal metrics with respect to deciding on research funding and the appointment and promotion of academics and researchers, and to assess instead the real value of their publications.

In the last years, the "publish or perish" syndrome has taken over worldwide [21–24], with a high negative impact on the research community, and especially on young researchers [5,18,25–27]. On the one hand, the researchers exploiting the new system (used for promotion and funding), often by speculative means, or by including already established scholars in teams with a lower profile, have formed a new "elite" and accumulated more and more money from research grants, while teams of young researchers have a very limited access to financial resources, although their ideas are very creative and useful for science [27].

Currently, since there are many discussions and contested criteria in the evaluation of individual researchers or academics and their institutions, different indexes or indicators should be used at the national level in order to capture the research landscape at the regional and global levels. In such analyses, the scholars usually take into consideration two databases, i.e., those offered by Clarivate Analytics and Scopus, and very rarely Google Scholar or Research Gate. Cavacini [28] shows, based on a study of the scientific output of Middle Eastern countries, that in the last years the number of regional analyses focused on Islamic countries, Latin America and the Baltic countries, which dominantly use Clarivate Analytics and Scopus data, has increased substantially.

The recent literature includes few dozens of articles published on different issues of research performance in Eastern European countries, using more or less scientometric indicators or indexes [6,10,12,23,29,30]. The dominant topics of these articles include comparative analyses between the Eastern European countries, between the Central and Eastern European ones [23], between these countries and different countries from Western Europe [31] or the evaluation of the impact of Brexit on the collaboration between the European Union (EU) countries [32]. An interesting study regarding the output of university research took into consideration the impact factor of journals (assessed based on the former Thomson Reuters database), starting from the idea that this indicator includes indirectly the number of citations [33]. However, the literature seems to lack studies carried out at a broader, regional scale, focusing on the recent administrative and political reforms in research and education, and on their impact on the publishing landscape.

In order to fill in this gap, the present study analyzes the research performance in the Central-Eastern European countries, taking into consideration 23 countries which belonged to the former communist area. The study focuses on the relative research performance, because it takes into consideration only a comparative analysis of the countries within this geographical area. The analyses are carried out in tight connection with the impact of the dependency on the current research performance to different extents, the historical East-West connections between some countries, the national research policies and the current attraction exercised by the new scientometric indicators on scholars and their use as a means of obtaining a better position in the different institutional or individual rankings.

## 2. Materials and Methods

The analysis uses only the information provided by Scopus, which includes a very large number of journals, highly appreciated by the majority of academic communities from these countries. The study used SCImago's country rank data, freely available from the Internet [34], covering the period 2010–2018, and the overall data for 1996–2018. The choice of Scopus against Clarivate Analytics was made because only Scopus has aggregated data for institutions and countries. The data set covers "Eastern Europe", more exactly Albania, Armenia, Azerbaijan, Belarus, Bosnia and Herzegovina, Bouvet Island, Bulgaria, Croatia, Czech Republic, Estonia, Georgia, Hungary, Latvia, Lithuania, Macedonia, Moldova, Montenegro, Poland, Romania, the Russian Federation, Serbia, Slovakia, Slovenia and Ukraine. Data on Bouvet Island were removed due to their scarcity. The data sets were downloaded in a Microsoft Excel 2007 or later format and converted to a Microsoft Excel 2003 or earlier format in order to carry out the data processing and analysis.

Central and Eastern European countries are very diversified by their history, demographic size, number of researchers, scientific outputs and national research politics, elements strongly reflected

in the differentiated dynamics of the number of citations per article. Quality was assessed through a double perspective: the scientific output and its visibility. The variable used to analyze the scientific output was the number of documents. We were unable to adjust it for other variables (such as the number of researchers) due to lacking data for many countries and years during the period of analysis. The indicator used to assess the visibility of scientific output was the number of citations per document, chosen due to its independence from the size of the scientific output and to the fact that the total number of citations hides the real impact of each article. In order to correctly assess the visibility, two effects were considered. The first one was the fact that the variation of the share of citable documents from the total number of documents might affect the number of citations per document, which is computed with respect to the total number of documents and not only to the citable ones. The second was the self-citation rate, computed as the share of self-citations from the total number of citations and self-citations; accounting for it, the number of citations per document was adjusted excluding the self-citations. Self-citations are defined by SCImago as "number of journal's self-citations in the selected year to its own documents published in the three previous years, i.e., self-citations in year X to documents published in years X-1, X-2 and X-3; all types of documents are considered." [35] The adjusted index was computed using the formula in Equation (1):

$$Adjusted\ no.citations/doc. = \frac{No.citations\ -\ No.self-citations}{No.citations} \times No.citations/doc. \tag{1}$$

The analysis revealed an effect of "small figures", consisting of the fact that countries with a low scientific output generate outliers more easily; in order to counter this fact, if a country published no articles in a given field and year, the values of all other indexes were filled with zeroes. Moreover, data were aggregated by wider research areas: basic sciences (chemistry; physics and astronomy; mathematics), earth and life sciences (agricultural and biological sciences; veterinary; biochemistry, genetics and molecular biology; immunology and microbiology; earth and planetary sciences; environmental science), social sciences (social sciences; business, management and accounting; economics, econometrics and finance), medical sciences (pharmacology, toxicology and pharmaceutics; medicine; neurosciences; nursing; psychology; health professions; dentistry), engineering (decision sciences; computer science; energy; engineering; chemical engineering; materials science), arts and humanities, multidisciplinary and all subjects together.

Since the performance was assessed with two indicators, the number of documents (measuring the scientific output) and the adjusted citation index (measuring the visibility), we analyzed their relationship by computing Pearson's coefficient of linear correlation for the individual research areas, aggregated domains and all fields together, by country and type domain (individual or aggregated). In addition, noticing differences between the ranks based on the scientific output and those based on visibility, we ran an ANOVA model using the difference as dependent variable and the year, country and domain (individual or aggregated) as independent variables.

A similar analysis of different major aggregated fields was published by Radosevic and York [36], using the Thomson Reuters (now Clarivate Analytics) database and indicator, with different goals: to individualize the dynamic of specializations belonging to all scientific areas of some large regions of the world (including Central and Eastern Europe). Another important study pertaining to this issue was authored by Fiala and Willett [37], focusing on the former communist bloc countries, and using two main indicators: the number of publications and citations in computer science; the results showed a similar structure and dynamics of sub-fields, although with important differences. In our study, in order to facilitate the comparisons, country ranks were computed for each index. However, the rank indicator is very simple and provides more information about the position and changes in the place of each country in a regional hierarchy. This information enables the possibility to analyze the trends in the regional context of each individual or aggregated field and perform comparative analysis.

In addition to comparisons at the level of the entire region, further detailed analyses were carried out for several countries, used as examples. We have selected them taking into account the position of

each one in the regional rankings, the number of journals indexed in Scopus and their dynamics and own research politics in correlation with their strengths by science branches. Apart from Romania, which represents the main focus of the article, two other countries were analyzed comparatively. Poland was chosen to illustrate the example of a large country, with a critical mass of researchers in all fields and strong research policies supported by appropriate funding. Slovenia is smaller than Romania, but more supportive to research and the knowledge-based economy, able to support and willing to fund research consistently. At the same time, Romanian scholars had fewer international connections, due to a strong isolation policy also affecting research, in comparison with scholars from Poland and Slovenia. However, Polish research policies were focused, during the communist times, on the cooperation with the COMECON countries, relaxing the scientific cooperation with some Western countries. Slovenia had a particular position, taking into account that the former Yugoslav communist regime has usually encouraged scientific connections with scholars from Western countries. By its geographical, historical and geopolitical position, Slovenia has benefited from consistent research facilities.

In order to explain the findings of our study, Eurostat's "Europe 2020 indicators - R&D and innovation" [38] and "Population and population change statistics" [39] were used; the selected indicators were the R&D expenditure, by sectors of performance (rd_e_gerdtot), R&D personnel – number of researchers per 1000 people, by sectors of performance (rd_p_persocc), and population (demo_gind). The indicators were used to compute correlations with the visibility of research (measured through the adjusted number of citations per document), using both ranked and raw data, and carry out a multiple regression analysis using the adjusted number of citations per document as a dependent variable and the indicators and their interaction as independent variables.

Since the last set of data did not include the values for Russia in 2013, 2015 and 2016, the values were filled in with Wikipedia data [40], since all other values were in line with the Eurostat data. Based on the data, a new indicator, the number of R&D personnel per 1000 people, was derived from the data on R&D personnel and population using the formula in Equation (2):

$$No. R\&D\ personnel/1000 people = \frac{No. R\&D\ personnel}{Population} \times 1000 \tag{2}$$

For all the countries with available Eurostat data (Bosnia and Herzegovina, Bulgaria, Croatia, Czech Republic, Estonia, Hungary, Latvia, Lithuania, Macedonia, Montenegro, Poland, Romania, Russian Federation, Serbia, Slovakia and Slovenia), a correlation analysis was run for the raw variables (adjusted number of citations per document, R&D expenditure and number of R&D personnel per 1000 people) and a multiple regression analysis was computed, using the adjusted number of citations per document as a dependent variable and the R&D expenditure, number of R&D personnel per 1000 people and their interaction as independent variables. The Statistical Package for the Social Sciences (SPSS) was used for both analyses.

In addition, the variables were re-ranked for the countries with available data, and correlations computed between the rank based on the adjusted number of citations per document and the ranks based on R&D expenditure, respectively, on the number of R&D personnel per 1000 people, using formulae implemented in Microsoft Excel 2003.

## 3. Results

### 3.1. Influence of a Critical Mass of Researchers and Research Expenditure

A first analysis, carried out at the level of the entire region, looked at the importance of the number of researchers and funding. The results of the correlation analysis are presented in Table 1 and those of the regression analysis in Table 2.

**Table 1.** Correlations between the adjusted number of citations per document and the R&D expenditure and number of researchers per 1000 people, using both ranked and raw data.

| | Data | Adjusted Number of Citations | | | R&D Expenditure | | | Number of Researchers per 1000 People | | |
|---|---|---|---|---|---|---|---|---|---|---|
| **Adjusted Number of Citations** | Ranked data | — | | | r 0.49 | P ≤0.001 | n 93 | r 0.58 | p ≤0.001 | n 88 |
| | Raw data | | | | r 0.29 | P 0.005 | n 95 | r 0.32 | p 0.002 | n 90 |
| **R&D Expenditure** | Ranked data | r 0.49 | P ≤0.001 | n 93 | — | | | r 0.71 | p ≤0.001 | n 81 |
| | Raw data | r 0.29 | P 0.005 | n 95 | | | | r 0.78 | p ≤0.001 | n 83 |
| **Number of Researchers per 1000 People** | Ranked data | r 0.58 | p ≤0.001 | n 88 | r 0.71 | p ≤0.001 | n 81 | — | | |
| | Raw data | r 0.32 | p 0.002 | n 90 | r 0.78 | p ≤0.001 | n 83 | | | |

**Table 2.** Multiple regression analysis using the adjusted number of citations per document as a dependent variable and the R&D expenditure, no. of R&D personnel per 1000 people and their interaction as independent variables.

| Tested Dependence of the Adjusted Number of Citations per Document | F | *p*-Value |
|---|---|---|
| Overall model | 5.54 | 0.02 |
| R&D expenditure | 6.18 | 0.02 |
| No. of R&D personnel per 1000 people | 2.22 | 0.14 |
| Interaction between the R&D expenditure and no. of R&D personnel per 1000 people | 5.00 | 0.03 |

The results of the correlation analysis indicate, in all cases, a significant positive influence of the R&D expenditure and the number of researchers per 1000 people on the adjusted number of citations per document. These findings are strengthened by the results of the regression analysis, which shows that the R&D expenditure, the number of researchers per 1000 people and their interaction can be used as an explanatory model for the variation of the adjusted number of citations per document. Although in the regression analysis the partial regression of the adjusted number of citations per document on the number of researchers per 1000 people was not found to be significant (F = 2.22, p = 0.14), given that the interaction of the R&D expenditure and the number of researchers per 1000 people was found to be significant (F = 5.00, p = 0.03), justifies the inclusion of the number of researchers per 1000 people in the regression model.

*3.2. Differentiated Dynamics of the Adjusted Citation Index*

In order to identify the trends and groups of countries based on their individual dynamics between 2010 and 2018, Figure 1 shows the specific position of each country, based on the rank in 2010 and the difference between the ranks in 2010 and 2018. A careful analysis of this graphic shows that some of the small countries born out of the fragmentation of the former Soviet Union (e.g., Armenia, Azerbaijan, Lithuania, Latvia or Belarus) and part of the countries belonging to the former Yugoslavia (Macedonia or Bosnia Herzegovina) rank higher in the 2018 hierarchy.

The diversity of the analyzed area is well highlighted by the gaps between countries with respect to the average values of the adjusted citation index during 1990–2018. The highest values, over eight citations per document, are found for small countries, three of them belonging to the former Soviet Union (Estonia, Georgia and Armenia), and the others are well-known for their performance in Central Europe (Slovenia, Hungary and Czech Republic). The opposite part of the hierarchy includes a mix of countries, grouping the largest (Ukraine and Russian Federation), medium (Romania, Poland,

Azerbaijan) and small ones (all the former Yugoslavian countries, except for Slovenia and some countries belonging to the former Soviet Union).

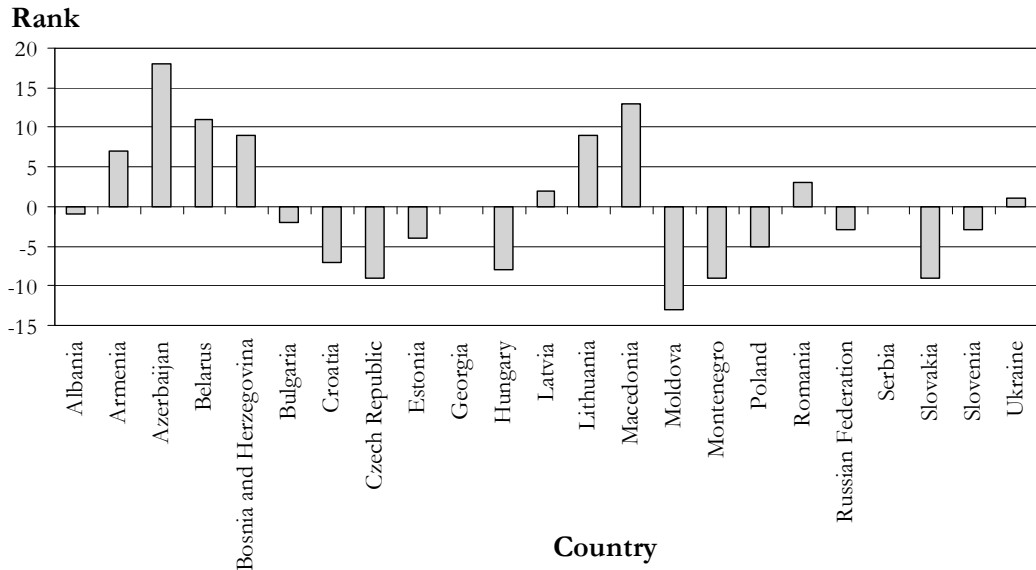

**Figure 1.** Dynamics of the differences between the rankings based on the adjusted citation index in 2010 and 2018 for the Central and Eastern European countries. The countries are displayed in ascending order regarding the ranks of the adjusted citation index in 2010.

In the last years these countries developed appropriate politics to encourage research at the individual and institutional level. The opposite part of the graphic includes countries with good results in publishing (Slovenia, Czech Republic, Slovakia, Hungary or Poland, which lost a few places) and small countries, which need to improve their scientific performance (Moldova, Montenegro), except for Estonia, which has a good research activity. The first group increased the number of published articles and, implicitly, of domestic journals indexed in Scopus. Due to this policy, for instance, the number of self-citations increased, and the values of adjusted citation index were also affected. The changes in the ranking dynamics for the other countries were very small; these countries occupy approximately the same places in the regional ranking.

### 3.3. Citable Documents and Self-Citation Effects

In a first stage, the analyses looked at a correct assessment of the visibility, accounting for the share of citable documents and self-citation effects. The analysis was performed globally, using the 1996–2018 data for all fields, since particular variations due to the status of a given field in a certain country would be harder to assess. Table 3 presents two important indicators. The first one is the citable/total documents ratio, important for the scientific output. The results show that there are slight variations between countries, but the values have a narrow range (0.91 to 0.98). However, the self-citation effect shows considerable variation: the share of self-citations ranges from 7.89% in Albania to 24.58% in the Russian Federation, requiring an adjustment of the number of citations per document. The differences seem to correlate to the country size, meaning that larger countries (Russian Federation, Ukraine and Poland) have higher self-citation rates than the smaller ones (Albania, Macedonia, Bosnia and Herzegovina, Latvia).

**Table 3.** Analysis of the effects due to citable documents and self-citations on the number of citations per document in Eastern European countries (1996–2018).

| Country | Citable/Total Documents Ratio (*) | Adjustment Based on the Share of Self-Citations (**) | Self-Citation Based Rank (***) |
|---|---|---|---|
| Albania | 0.94 | 7.89 | 23 |
| Armenia | 0.97 | 15.51 | 12 |
| Azerbaijan | 0.97 | 15.67 | 11 |
| Belarus | 0.98 | 15.76 | 10 |
| Bosnia and Herzegovina | 0.94 | 10.19 | 21 |
| Bulgaria | 0.96 | 12.42 | 19 |
| Croatia | 0.95 | 15.22 | 13 |
| Czech Republic | 0.96 | 17.82 | 5 |
| Estonia | 0.94 | 12.79 | 16 |
| Georgia | 0.91 | 10.28 | 20 |
| Hungary | 0.94 | 12.94 | 15 |
| Latvia | 0.96 | 12.62 | 17 |
| Lithuania | 0.97 | 15.85 | 9 |
| Macedonia | 0.95 | 8.73 | 22 |
| Moldova | 0.96 | 12.51 | 18 |
| Montenegro | 0.94 | 16.74 | 6 |
| Poland | 0.96 | 20.14 | 3 |
| Romania | 0.96 | 18.34 | 4 |
| Russian Federation | 0.98 | 24.58 | 1 |
| Serbia | 0.94 | 16.34 | 7 |
| Slovakia | 0.97 | 15.95 | 8 |
| Slovenia | 0.95 | 13.63 | 14 |
| Ukraine | 0.98 | 21.27 | 2 |

* Share of citable documents from the total number of documents; ** 100 × number of self-citations/(total number of citations + number of self-citations): the higher its value, the higher the self-citation rate; *** ranking based on the adjustment based on the self-citation effect: the lower the values, the lower the self-citation effect.

*3.4. Analyses of the Dynamic of Individual and Aggregated Fields*

An overview of the dynamics of research performance, analyzed using the citation adjusted index, demonstrates different patterns, mirroring specific conditions, traditions and policies. For this reason, the results concerning each field will not be presented in detail, but some examples are illustrative:

- A dramatic change of ranks can be noticed for different fields in different years: multidisciplinary research (2014 and 2017), arts and humanities (2012 and 2017), computer sciences (2013 and 2015), dentistry (2015 and 2017), health professions (2015 and 2017), economics, econometrics and finance (2013 and 2017) and business, management and accounting (2011 and 2016); in these years, countries that occupied the top positions ranked very low, but returned to their previous position immediately afterwards. As can be seen, there is no particular year when these dramatic changes occur, and even for related fields, such as economics, econometrics and finance and business, management and accounting, the shift occurs in different years (2015 and 2017, and 2013 and 2017, respectively).

- Less dramatic changes of ranks occur all over the time. In general, smaller countries show less consistent patterns than the larger ones, most likely due to the "small figures" effect, consisting of the fact that smaller figures are more likely to result into disproportionate ratios than the larger ones. For example, in 2011 Albania published 2 articles on dentistry, but received 163 citations of earlier articles, situating it at the top with 81.5 citations per article.

The Analysis of Two Main Fields: Dentistry and Engineering Sciences

For some fields, the ranks of the analyzed period (2010–2018) are different from the overall ranks (computed for the period 1996–2018); this statement will be illustrated in more detail in the analysis

of individual countries. In general, the analysis revealed inconsistent patterns and changing ranks, especially when looking at given fields and certain years.

In order to illustrate this statement, Figure 2 shows the rank variation for dentistry science. The overall picture of its dynamics shows real chaotic trajectories for the entire Central and Eastern European space, with strongest fluctuations in 2015. This means that the research activity in this field is not a constant preoccupation of researchers, which explains the differences from one year to another for each country. Two countries show fewer fluctuations: Slovenia, Hungary and Lithuania; all others drop or increase by 10 places in only one year. Croatia shows some linear dynamic, with ranks between 10 and 14 each year. A very interesting case study is Albania, which, due to the very few articles, presents the highest fluctuations: in two years (2011 and 2015), it seems to be the best performer, and in the three years between (2012, 2013 and 2014), the poorest one.

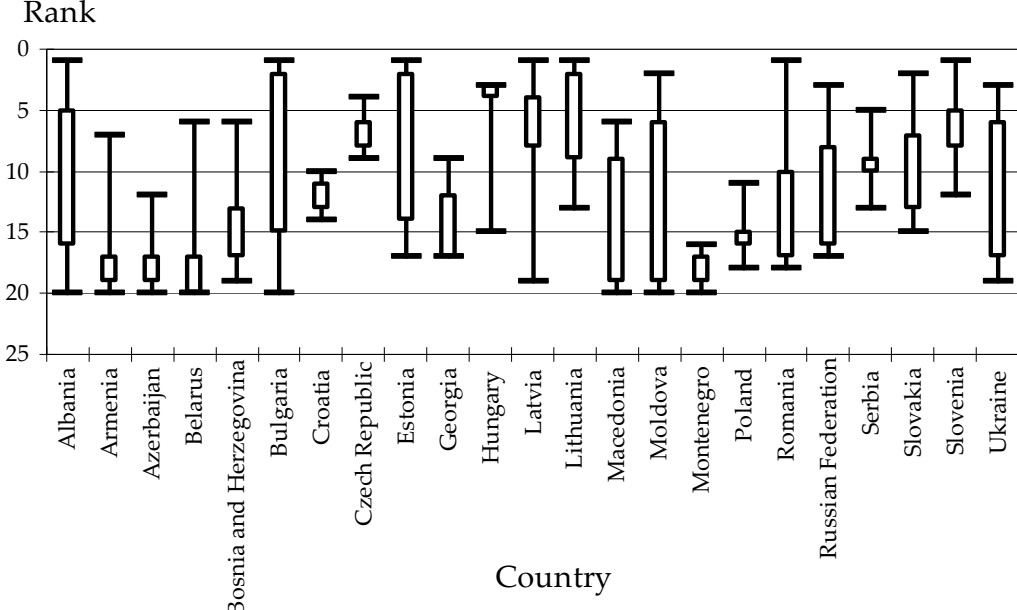

**Figure 2.** Showing the visibility of articles published on "dentistry" in Eastern European countries during 2010–2018, measured based on the adjusted number of citations per document. The figure displays the ranks of countries computed for each year separately. Results indicate a huge variability and inconsistent patterns for most of the countries analyzed.

In opposition to the dentistry field, the dynamic of engineering sciences (Figure 3) indicates stronger profiles, reflected in the constant rank distribution of several groups of countries. The effects of small countries with few articles but many citations affect the hierarchy, especially after 2015. In Figure 3, the dominance of the Republic of North Macedonia and Armenia during the last part of the analyzed interval, both unknown at the level of spectacular research in the engineering field, is an interesting result. The findings indicate a fluctuating overall picture. However, it shows the first consistent trend, revealed by the aggregated analysis: high-ranking countries exhibit variations with lower ranges and, in general, have steadier dynamics.

For some aggregated research areas, the ranks of the analyzed period (2010–2018) are different from the overall ranks (computed for the period 1996–2018); this statement will be illustrated in more detail in the analysis of individual countries.

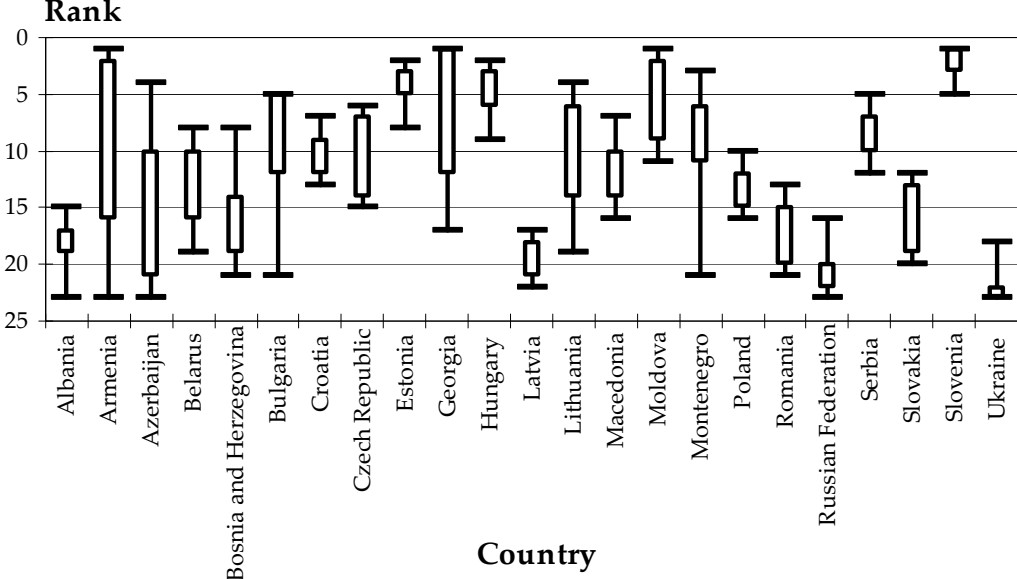

**Figure 3.** The visibility of articles published on aggregated "engineering" sciences (including decision sciences; computer science; energy; engineering; chemical engineering; materials science) in Eastern European countries during 2010–2018, measured based on the adjusted number of citations per document. The figure displays the ranks of countries computed for each year separately. Results indicate a huge variability and inconsistent patterns for most of the countries analyzed, except for the occupants of top positions, which exhibit steadier dynamics.

### 3.5. Analysis of the Relationships between the Scientific Output and Its Visibility

Table 4 displays partial results of the correlation analysis using the number of documents as a measure of the scientific output and the adjusted citation index as a measure of its visibility. The results were inconclusive; overall, for each country and each domain, both positive and negative results were identified as being statistically significant; they are displayed in the table.

**Table 4.** Analysis of the correlations between the number of documents and the adjusted citation index in the Eastern European countries (1996–2018). The table displays only the statistically significant correlations at two levels ($\alpha = 0.05$ and $\alpha = 0.1$).

| | Type of Analysis | | | | | |
|---|---|---|---|---|---|---|
| | **Individual Research Domains** | | **Aggregated Fields** | | **All Fields** | |
| **Correlation [1]** | **+** | **—** | **+** | **—** | **+** | **—** |
| **Overall** | Yes | | | Yes | | Yes |
| **Country** | $0.1 \leq p < 0.05$: Armenia; $p \leq 0.05$: Moldova, Montenegro | $0.1 \leq p < 0.05$: Serbia; $p \leq 0.05$: Belarus, Bulgaria, Croatia, Hungary, Latvia, Lithuania, Poland, Romania, Russian Federation, Slovakia, Slovenia, Ukraine | $0.1 \leq p < 0.05$: Albania; $p \leq 0.05$: Bulgaria, Croatia, Czech Republic, Estonia, Georgia, Latvia, Lithuania, Poland, Romania, Russian Federation, Serbia, Slovakia, Slovenia, Ukraine | | $p \leq 0.05$: Bosnia and Herzegovina, Croatia | $p \leq 0.05$: Albania, Slovakia |
| **Domain** | $0.1 \leq p < 0.05$: agricultural and biological sciences; $p \leq 0.05$: decision sciences, computer sciences, chemical engineering, neurosciences, nursing, psychology, health sciences, dentistry | $p \leq 0.05$: engineering, medicine | $p \leq 0.05$: engineering, medical sciences, social sciences | | | |

"[1]","+" indicates positive correlations, and "—" negative correlations.

The analysis of variance looking at the dependence of the difference between the ranks based on the number of documents and those based on the adjusted citation index indicated that its dependence on the country is significant, but did not find sufficient evidence for its dependence on the year and domain.

### 3.6. Three Case Studies: Poland, Slovenia and Romania

### 3.6.1. Poland

Scientific Output

The analysis of the scientific output in Poland during 1996–2018 is displayed in Figure 4 for all fields and in Figure 5 for the aggregated ones. The results rank Poland in the top five of the ranking for individual research areas, and in the middle of the ranking for the aggregated fields. Poland seems to rank worse in medicine, decision sciences and business, management and accounting among the individual areas, while the performance of the aggregated fields is more homogeneous.

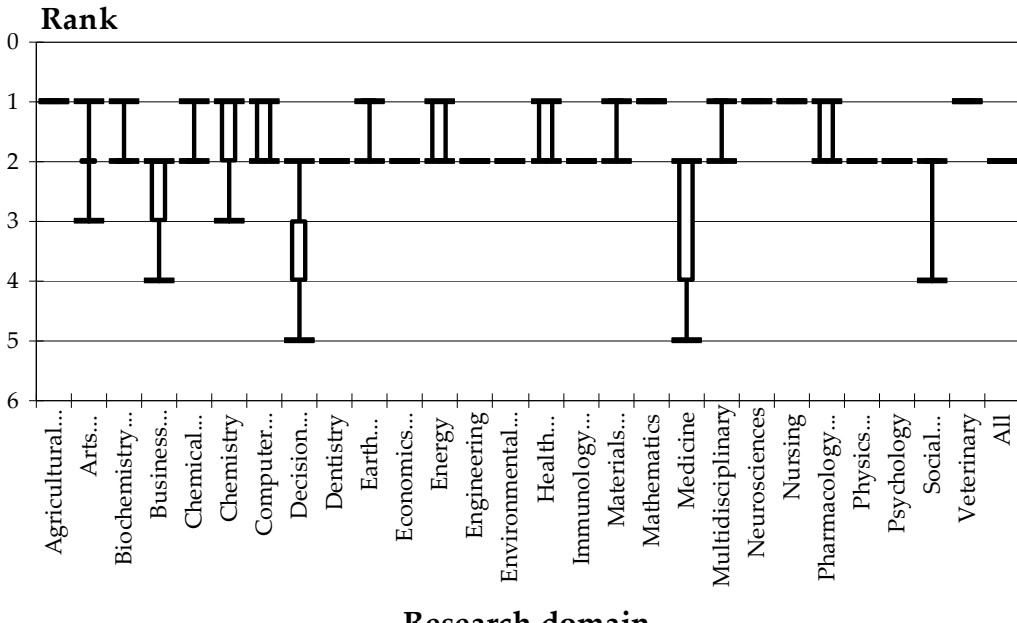

**Figure 4.** The number of documents published in different fields in Poland during 2010–2018. The figure displays boxplots of the ranks of fields computed for each year separately. Results indicate that the values are in the ranking's top five.

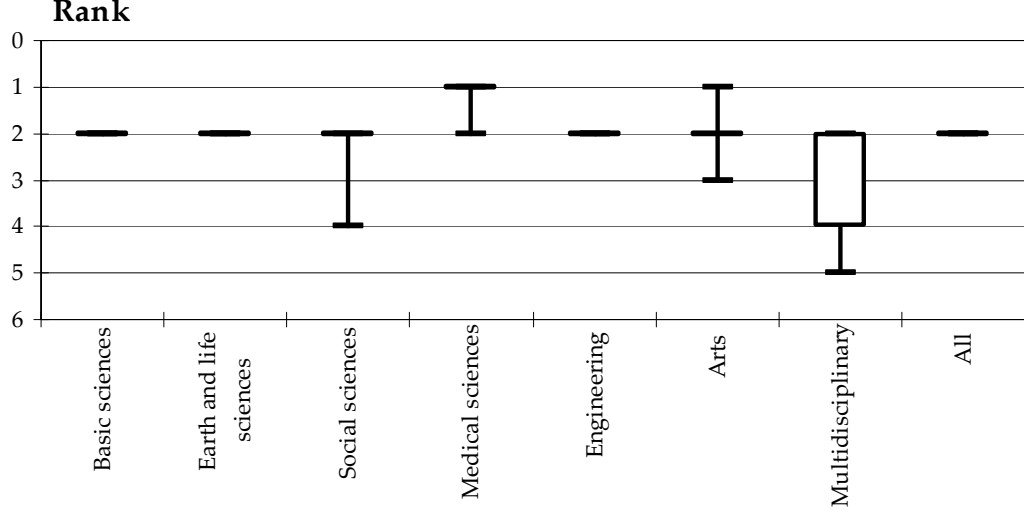

**Figure 5.** The number of documents published in different aggregated research areas in Poland during 2010–2018, measured based on the adjusted number of citations per document. The figure displays boxplots of the ranks of fields computed for each year separately. Results indicate that the ranks cluster in the first quarter of the ranking.

Visibility

The analysis of individual fields (Figure 6) revealed a huge variation, with shifts of ranks in 2012, 2014 and 2017. The overall picture shows a great variability, with ranks ranging between 3 and 20, although for most fields the ranks are situated between 8 and 14. The least variable fields seem to be earth and planetary sciences (9–13), materials science (10–14) and mathematics (12–16).

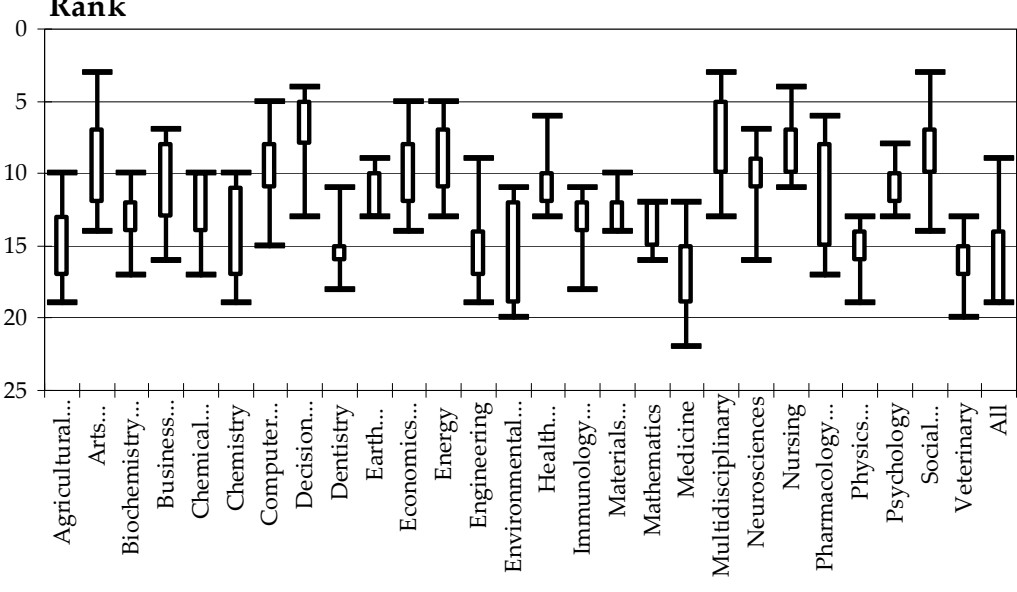

**Figure 6.** The visibility of articles published in different fields in Poland during 2010–2018, measured based on the adjusted number of citations per document. The figure displays boxplots based on the ranks of the fields computed for each year separately. Results indicate a huge variability and an "average" performance, with relatively steady trends.

The analysis of aggregated research areas revealed, again, a great variability, with ranks ranging from 3 to 19, but in this case some consistent patterns could be identified (Figure 7).

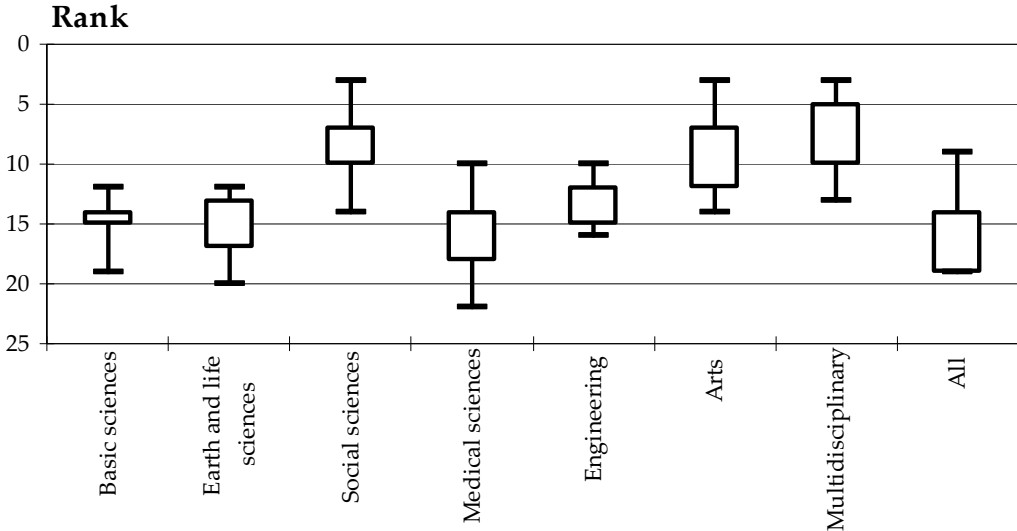

**Figure 7.** The visibility of articles published in aggregated research areas in Poland during 2010–2018, measured based on the adjusted number of citations per document. The figure displays boxplots based on the ranks of the aggregated research areas computed for each year separately. Results indicate a huge variability and allow for the identification of several patterns.

Looking at these differences, two trends can be easily discerned: health sciences (nursing, psychology, health professions and dentistry) and veterinary (which, although included in the aggregated "earth and life sciences", is also related to health) seemed to be doing better in the analyzed period compared to their overall rank. On the contrary, the other fields exhibited lower ranks during the analyzed period compared to their overall situation.

The analysis of aggregated domains also shows variations: for basic sciences, ranks ranged between 12 and 18 during 2010–2018, while the overall ranking was 11; for engineering, the ranks varied between 10 and 16 in the analyzed period (2010–2018), whilst the rank for the 1996–2018 was 10. This is consistent with the previous set of findings, reconfirming the decline during the study period.

3.6.2. Slovenia

Scientific Output

The analysis of the scientific output in Slovenia during 1996–2018 is displayed in Figure 8 for all fields and in Figure 9 for the aggregated ones. The status of Slovenia seems to be diametrically opposed to that of Poland. For most individual research areas, Slovenia is positioned in the middle of the ranking, but situated in the first part of the ranking of the aggregated areas with a large variability (earth and life sciences, medical sciences and multidisciplinary domains do not perform as well as the others).

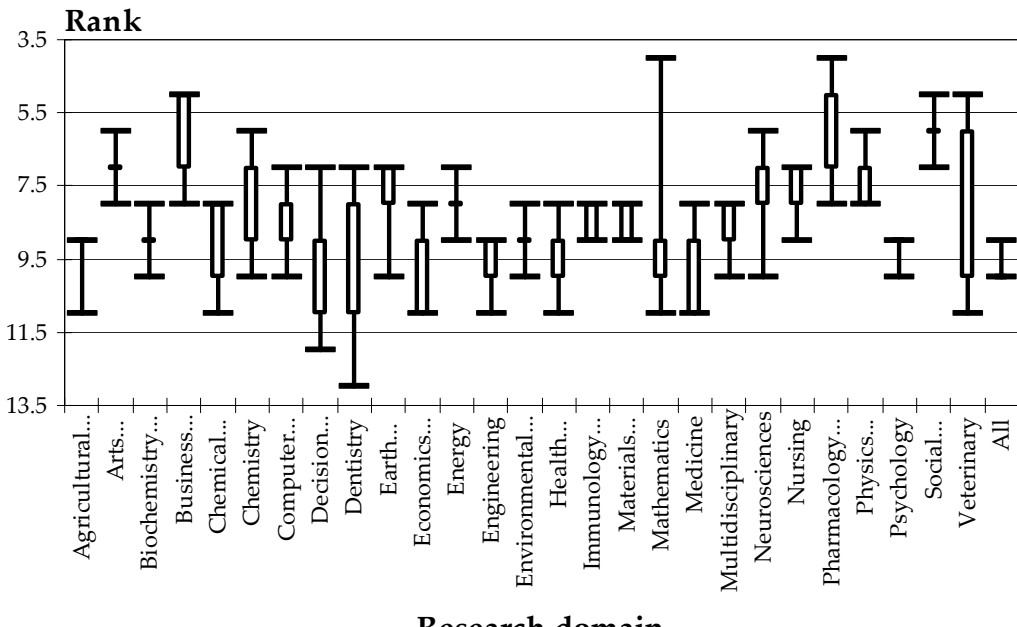

**Figure 8.** The number of documents published in different fields in Slovenia during 2010–2018. The figure displays boxplots of the ranks of fields computed for each year separately. Results indicate that the values are in the middle part of the ranking.

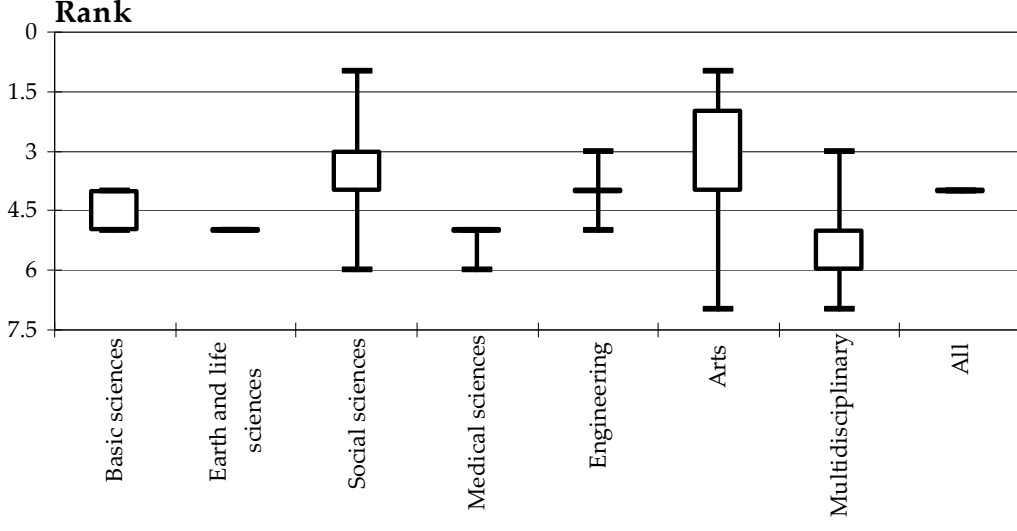

**Figure 9.** The number of documents published in different aggregated research areas in Slovenia during 2010–2018, measured based on the adjusted number of citations per document. The figure displays boxplots of the ranks of fields computed for each year separately. Results indicate that the ranks cluster in the first third of the ranking, with a large variability.

Visibility

Slovenia appears to be a "top player" with respect to individual fields and aggregated research areas. For the individual fields, ranks ranged between 1 and 21, but most were in fact between 1 and 9 (top 1/3) (Figure 10). For mathematics and chemical engineering, ranks ranged between 1 and 4 overall, as well in the study period. The only domain where Slovenia did not thrive is the field of arts and humanities, where ranks ranged between 13 and 20 during 2010-2018 and the overall rank for 1996–2018 was 16. In social sciences, the ranks ranged from 9 to 19 during 2010–2018, while the overall rank

was 8; in nursing, ranks were between 3 and 16 during 2010-2018, and the overall rank was 10; these areas are the next ones where Slovenia has a weaker position, with a trend of improvement in nursing and a decreasing one in the social sciences (seen by comparing the overall ranking with the yearly ones during 2010–2018). The explanation for this performance be Slovenia's "smart specialization" in innovation and high tech, demanded by engineering fields since the communist times. In the last decades, this trend continued, especially in computer sciences, chemical industry, biochemistry, physics and others.

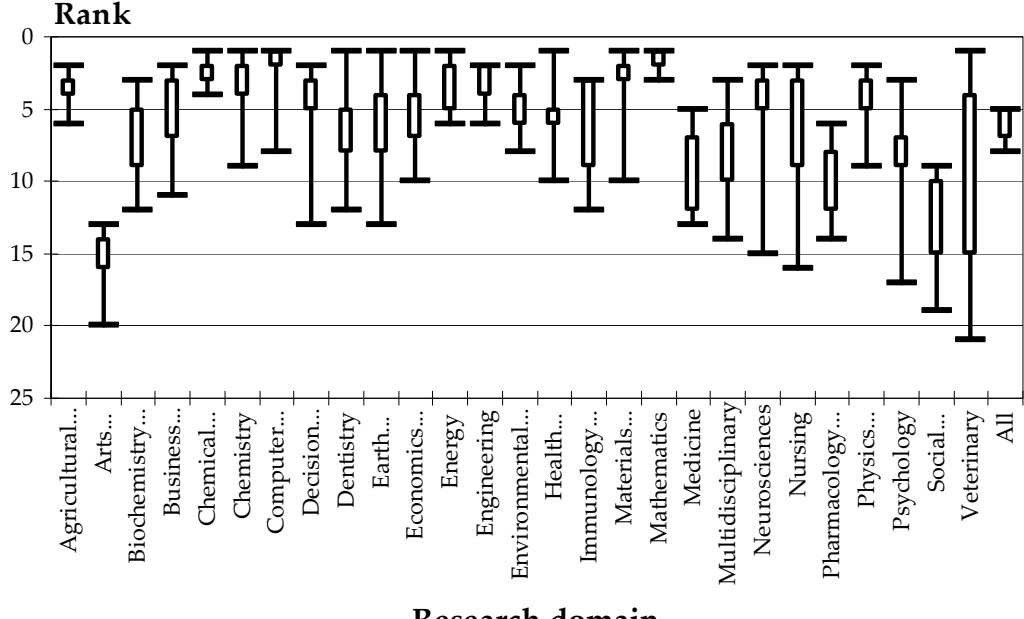

**Figure 10.** The visibility of articles published in different fields in Slovenia during 2010–2018, measured based on the adjusted number of citations per document. The figure displays boxplots based on the ranks of the fields computed for each year separately. Results indicate a huge variability and pinpoint a poorer performance in arts and humanities and social sciences.

Unlike Poland, Slovenia shows steadier trends when comparing the 2010–2018 visibility with the overall (1996–2018) one, with the exceptions presented in detail below, all suggesting a poorer performance during the analyzed period (2010–2018) compared to the overall status (1996–2018). For example, immunology and microbiology exhibited ranks between 3 and 12, whilst the overall rank was 1; the ranks for computer sciences varied between 1 and 8, whilst the overall rank was 1; environmental sciences showed ranks between 2 and 8, whilst the overall rank was 1; materials sciences exhibited ranks between 1 and 10, whilst the overall rank was 1; mathematics ranked between 1 and 3, whilst the overall rank was 1, etc. Overall, the findings are consistent and suggest that these domains fared worse in the study period than overall.

The results of the analysis of aggregated research areas are presented in Figure 11. The image shows the variable performance of many domains, although most ranks range between 1 and 9. The clearly poorer performance of arts is also visible. Moreover, the ranks for social sciences varied between 9 and 19 during 2010–2018, while the overall ranking for 1996–2018 was 8, suggesting a poorer performance. On the opposite side, a better performance was showed by engineering, with ranks between 1 and 5 and an overall ranking of 1, and overall, where the ranks varied between 5 and 8 during 2010-2018 and the overall ranking was 5. Last but not least, earth and life sciences, engineering and basic sciences presented steadier dynamics.

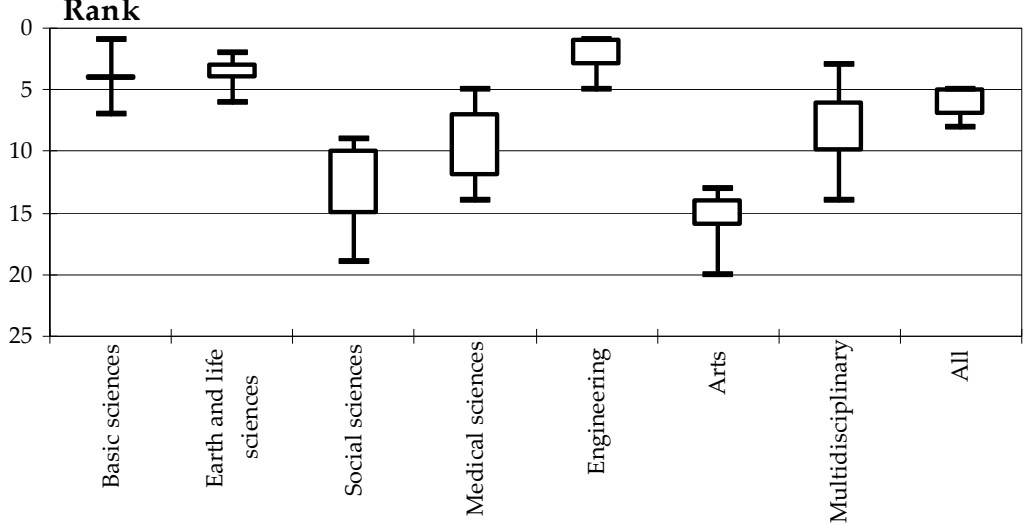

**Figure 11.** The visibility of articles published in aggregated research areas in Slovenia during 2010–2018, measured based on the adjusted number of citations per document. The figure displays boxplots based on the ranks of the aggregated research areas computed for each year separately. The results indicate variability across the research areas and a poorer performance in arts.

### 3.6.3. Romania

Scientific Output

The analysis of the scientific output in Romania during 1996–2018 is displayed in Figure 12 for all fields and in Figure 13 for the aggregated ones. Among the individual research areas, the performance is lower in mathematics, neurosciences, and agricultural and biological sciences, while the multidisciplinary domains are the poor performers among the aggregated research fields.

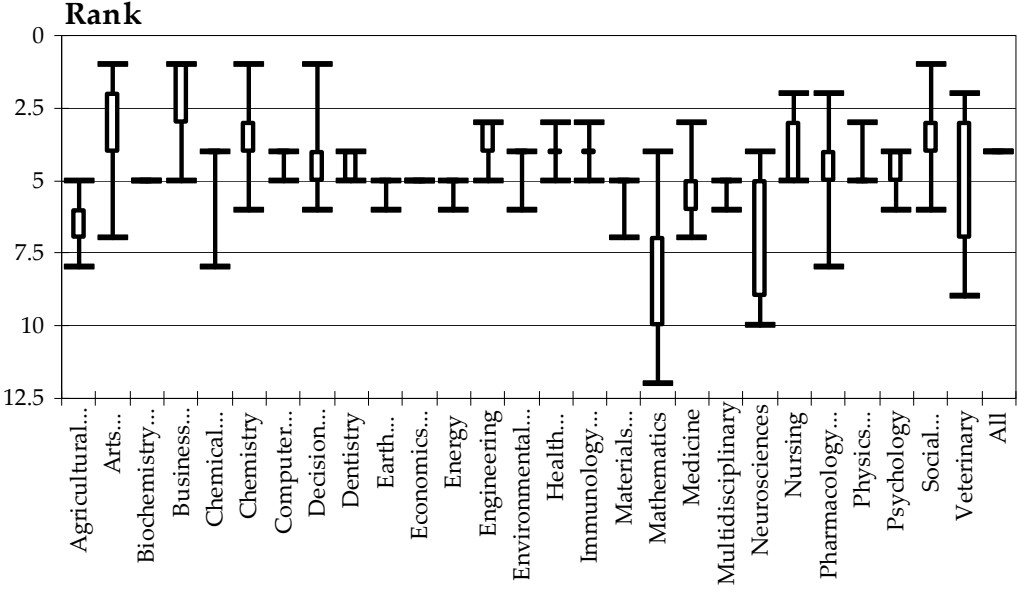

**Figure 12.** The number of documents published in different fields in Romania during 2010–2018. The figure displays boxplots of the ranks of fields computed for each year separately. Results indicate that the values are in the top part of the ranking for most fields.

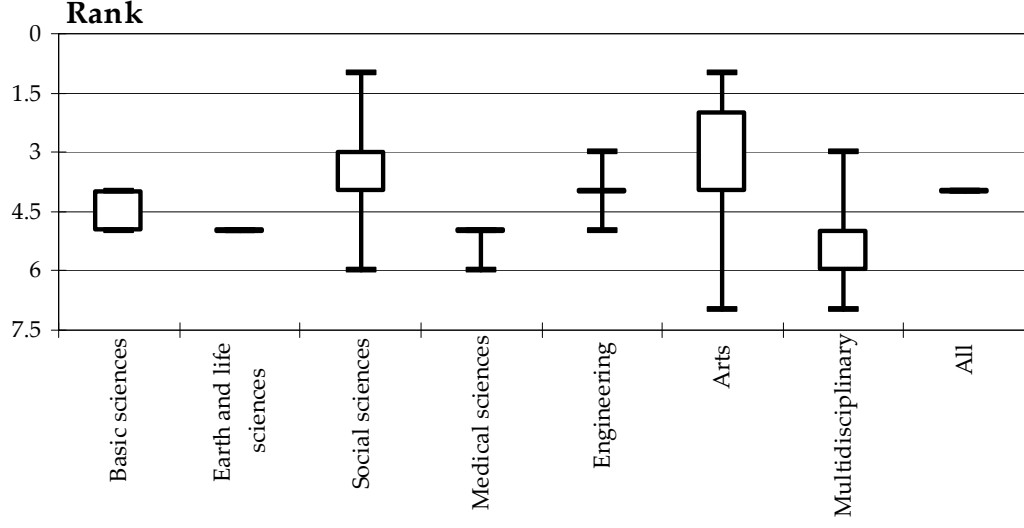

**Figure 13.** The number of documents published in different aggregated research areas in Romania during 2010–2018, measured based on the adjusted number of citations per document. The figure displays boxplots of the ranks of fields computed for each year separately. Results indicate that the ranks cluster in the first third of the ranking.

Visibility

If Slovenia was a "top player", Romania appears to be at the lower end of the "top". The analysis of the individual fields (Figure 14) reveals overall ranks situated between 17 and 22 for all fields, except for chemical engineering (8), mathematics (13) and computer sciences (13). The better performance is explained by the tradition of research: during the communist times, the dictator's wife received a debated doctoral degree in chemical engineering, and the domain was under her patronage. Ungureanu and Mincan [41] have published an interview with the President of the Romanian Academy at that time (a chemistry researcher), who explained how she earned her PhD degree and became a "world-renowned scientist" without meeting the academic standards. At the same time, their daughter worked for the Institute of Mathematics, which had a longer tradition; unlike her parents, her work was real. Computer sciences were stimulated in the post-communist times, although the roots existed even before.

There are also shifts in the ranks during 2010–2018 compared to the overall period covered by the data (1996–2018), all suggesting an improvement during the analyzed period (2010–2018) compared to the overall status (1996–2018). Veterinary science had ranks between 2 and 15, whilst the overall rank was 16; neurosciences ranked between 8 and 18, whilst the overall rank was 19; nursing ranked vary between 1 and 18, whilst the overall rank was 18; and health professions' ranks fluctuated between 4 and 17, whilst the overall rank was 17. Nevertheless, these important changes only had a sensible influence on the general position of Romania in the regional ranking of the Central and Eastern European countries. Overall, the findings are consistent and suggest that these domains fared better in the study period than overall.

The results of the analysis of aggregated research areas are presented in Figure 15. The findings indicate a poor overall performance, despite the recent trend. All domains fluctuated, except for social sciences, which exhibited an inexplicable ascending trend, probably an "accident". Another ascending trend is illustrated by the differences between the values between 2000 and 2018 and the overall ranking for 1996–2018: earth sciences showed ranks between 11 and 19, while the overall rank was 20, and the rankings for the arts varied between 12 and 21, while the overall ranking was 21.

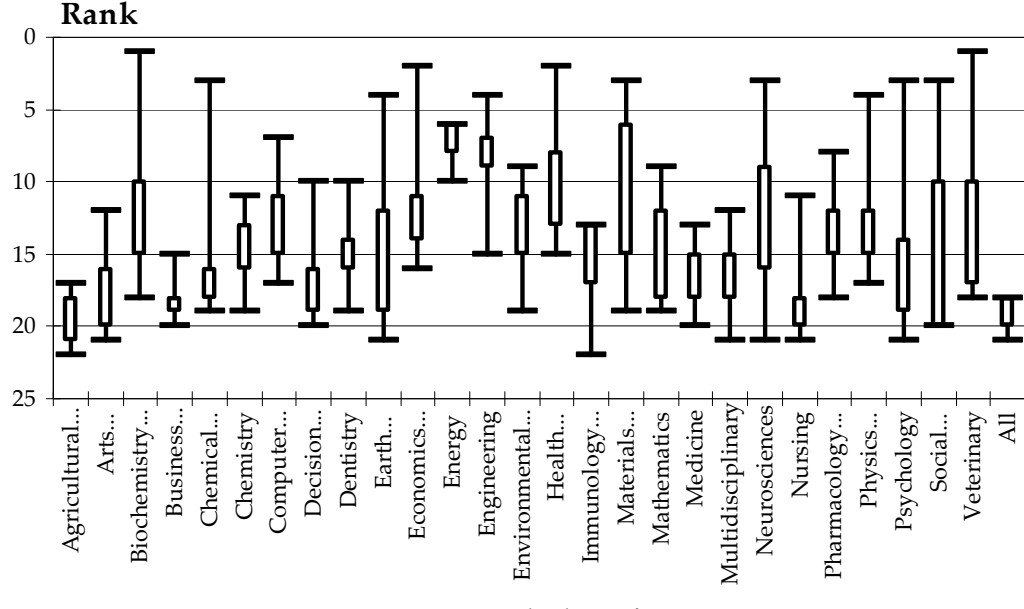

**Figure 14.** The visibility of articles published in different fields in Romania during 2010–2018, measured based on the adjusted number of citations per document. The figure displays boxplots based on the ranks of the fields computed for each year separately. Results indicate a huge variability and pinpoint an overall poorer performance, with important fluctuations.

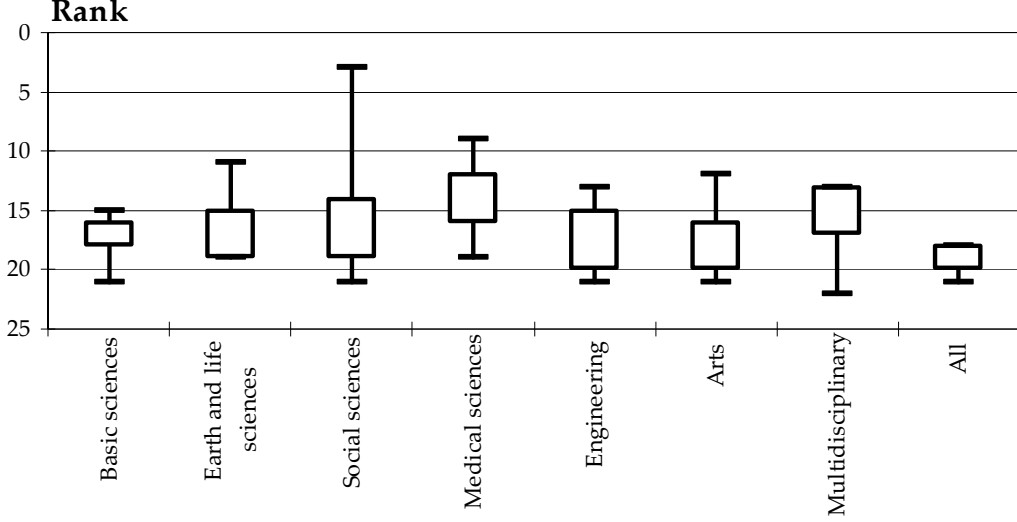

**Figure 15.** The visibility of articles published in aggregated research areas in Romania during 2010–2018, measured based on the adjusted number of citations per document. The figure displays boxplots based on the ranks of the aggregated research areas computed for each year separately. The results indicate the variability across the research areas and a poorer overall performance.

## 4. Discussion

### 4.1. Overall Analyses on the Influence of Financial Support

According to the recent literature, financial support is a crucial determinant for researchers and their institutions [42]. The GDP size and structure (especially R&D) are direct and indirect core indicators to appreciate the premises for sustaining the research and its performance [43]. A

comparative study made by Vinkler [44] shows that the relationship between GDP and the publishing indicators is very controversial. A direct relation with the number of publications does not explain the differences between Central and Eastern European countries with a low level of GDP and other, very developed countries. At the same time, some authors consider that the rankings of research and higher education institutions have an important influence on the expenditure on researchers and students. The competition for resources is a core element, and the last years demonstrated that the research institutions which applied more for grants and research contracts became more productive [45].

The financing system is very important in the national and international context. For the Central and East European countries, the facilities offered by the EU can be a key factor to valorize the creative potential of scholars from the entire continent. The new European mechanisms, starting from the idea that the financing system should be exclusively based on performance [46], could pose some problems regarding the limited access of some countries within this area. A good example is represented by the current EU condition that all research consortia must include a Central and Eastern European country (e.g., former FP7 Program or Horizon 2020). At the same time, there are some special programs for bilateral cooperation with other non-UE states, which create the premises for a better future cooperation. However, there are main obstacles, due to the history of each country and the reduced resources needed to finance the access to relevant journals, modernize the research infrastructure, improve communication, etc.

Although the relationship between GDP and research performance is contested in the literature, this study offers additional evidence supporting the importance of the R&D expenditures of each country for improving its research performance. Overall, the results indicate that sound policies for increasing the visibility of research require sufficient budget allocations and a certain status of the researcher, drawing people to the research activities in order to obtain a "critical mass" of researchers. Furthermore, the two actions must be taken in conjunction, to ensure their interaction.

### 4.2. The Scientific Output

In this research the self-citation phenomenon proved to be sufficiently important to determine us to adjust the visibility indicator used by SCImago, along with the number of citations, in order to account for it. The results presented in Table 3 and Section 3.3 indicate that, although SCImago provides data showing that not all documents are citable, the share of citable documents from all documents had a very little variation across Eastern Europe, with values ranging between 0.91 and 0.98. For this reason, we consider that the total number of documents can be used to measure the scientific output without having to adjust it in order to account for the fact that not all documents are citable. However, there are other possible adjustments, further discussed in the section dealing with the limitations of this study.

### 4.3. Self-Citations and their Effects

The self-citation phenomenon requires an in-depth discussion. Aggregated data are not sufficient for supporting judgments as to whether countries have national policies or a specific local research culture for dealing with them. The list of other elements with a potential influence includes: (1) the share of research fields where self-citations are more frequent (e.g., fields requiring local studies for the validation of results, due to social and cultural specificities—socio-economic sciences, or geographic particularities—earth sciences vs. fundamental research) or with specific citation patterns—e.g., in socio-economic disciplines, articles are not the main research outputs [47], and often studies cite documents other than the articles [48]; (2) language barriers: in many Eastern European countries, where English is not even the first foreign language, researchers often write an article in their own language, including a literature review based on the national literature published in their own language, and have it translated using specialized services; (3) Al-Suqri and Lillard [49] and Petrişor [50] show that research from developing countries often addresses local and national issues, including the literature review; (4) the impact of choosing the Hirsch index as a measure of individual research

performance; especially in the Central and Eastern European countries, the index played an important role in the promotion of researchers [19,27].

Usually, the Hirsch index is computed including the self-citations (although Scopus provides a version excluding them). The place of this index in assessing the research performance justifies why the countries with many scholars are situated in the first places of the hierarchy (Russian Federation, Ukraine, Poland, Romania or Czech Republic). This is the result of self-citations, frequently in newly indexed journals or in indexed conference proceedings.

### 4.4. Dynamic of the Individual and Aggregated Research Areas

Due to the fact that the analysis of individual fields was more affected by the small figures effect, given the differences in the number of articles published in different fields, new analyses were carried out for the aggregated research areas: basic sciences, earth and life sciences, social sciences, medical sciences, engineering, arts and humanities, multidisciplinary and all subjects together. Despite the aggregation, the results were again inconsistent, although less variable. Important rank shifts were present for the social sciences and the earth and life sciences in 2015 and 2017, but fluctuations were present for all aggregated research areas during the entire period. Looking at the dynamic of hierarchies, three dominant groups can be distinguished: (1) countries with a constant high adjusted citation index, (2) countries with constant rankings, but very low values, and (3) countries with spectacular dynamics or decline between 2010 and 2018. The first group includes three different countries, where Slovenia stands out by its constant top position in the yearly hierarchy (first place for three years, second for three and third for two). Estonia and Hungary usually kept their place in the first half of the Central and Eastern European hierarchy. The second group includes the largest (Ukraine, Russia, Romania and Poland) and smallest (Albania, Latvia and Slovakia) countries. If some are characterized by a constant presence in the last places (Ukraine and Russia, for example), others have declining ranks (Poland and Slovakia). The third group includes all other countries and comprises two categories: one with a serious decline and another with a spectacular increase. The first category includes countries with a serious decline in the first years (like Armenia) or entirely (Bulgaria). The second group has three countries with a spectacular growth: Armenia, Georgia (abrupt increase) and Lithuania. These countries are former Soviet republics, without a clear policy on the engineering research, except for Lithuania, which promotes high standards of higher education and research.

### 4.5. Relationships between the Scientific Output and Its Visibility

In relation to this issue, the results were surprising and inconclusive. The small sample size (i.e., low number of publications or citations for a given research area in a certain year and a specific country) might explain why not all correlations were found significant, but a question remains with respect to the sign of the relationship. Both positive and negative correlations were found to be statistically significant in certain countries or specific domains. Obviously, the explanations exist, but are not easy to find. Deeper analyses looking at the specific domain-related policies of each country analyzed, depending on their specific period of implementation, are most likely to constitute an explanation. It is interesting to see that the three countries analyzed in detail, although different in many other respects, showed a consistent pattern, meaning that this relationship was significantly negative for both the individual and aggregated domains, and not significant for all fields together.

### 4.6. Brief Analysis of the Case Studies

#### 4.6.1. Poland

With respect to the scientific output, Poland ranked highly. While it could be argued that the size of the country plays an important role, the other examples show that this supposition is not necessarily true. Despite the variability, Poland seems to be situated, in most fields, at the middle of the "visibility top". Most aggregated research areas suggest a descending trend (see, for example, engineering), with

a transitory tendency to revert starting in 2016 for the basic sciences and arts. Moreover, Poland seems to do better in social sciences, arts and multidisciplinary research than in the other fields. The analyzed period can also be characterized by the fact that particular fields showed a different dynamic during 2010-2018 compared to the overall period covered by the data (1996-2018). For some of them, the trends suggest an improvement (veterinary, material sciences, nursing, psychology, or health sciences), while for others the trend is reversed, suggesting a poorer performance (energy, biochemistry, mathematics, decision sciences, physics or earth sciences).

### 4.6.2. Slovenia

Slovenia has an average scientific output. Overall, the analysis of the visibility of Slovenia's results indicates that in the case of "top players" the variability across different research fields, aggregated or not, is lower, and their dynamics more predictable. At the same time, Slovenia and Poland show contrasting patterns: social sciences and arts perform better in Poland and worse in Slovenia. The recent apparently negative trends do not mean a poor performance of some sciences, because the number of publications from Slovenia continues to increase. The explanation is that other countries with an important scientific potential are more visible in the regional rankings. Slovenia inherited a very good research infrastructure and fruitful, cooperative relationships with scientists from Western countries (especially Austria), and it was one step ahead of the research reform started in the 1990s.

### 4.6.3. Romania

Important changes in Romanian research, especially within the universities, were promoted by the Education Law no. 1/2011 and other Ministerial Orders starting in 2016. Their expectations were very high regarding a better visibility of Romanian research in the world, by defining and enforcing new standards for assessing the institutions and individual researchers. The results of these policies are likely to explain why Romania, in terms of the scientific output, fares better than Slovenia, where researchers started their cooperation with the Western partners ten years earlier, under the framework of different European programs, and even better than Poland, despite the latter's size.

However, in visibility terms, the result is a sensible trend in the dynamics of ranks, especially in the aggregated research areas, which would consolidate in the next years at the regional level. The outstanding growth of social sciences could be accidental or indicate a new and spectacular progress in this field. Future research will either confirm this new trend or show that it was an exception.

Another important factor explaining the position of Romanian research reflected by Scopus is the fact that almost all standards and criteria used to assess the research outputs after 2011 take into consideration only the visibility given by journals indexed by Clarivate Analytics. Most of these journals are indexed in Scopus too, but due to the new criteria Romanian researchers are interested in publishing mainly in the journals included in the Web of Science. As a result, some end up by associating with predatory journals, as authors or part of their editorial boards [51–53].

The appearance of new categories of journals included in the Web of Science (i.e., the Emerging Sources Citation Index), which drastically reduced the gap between submission and publication [54], increased the attraction exercised by such journals on researchers. Normally, the peer review process is "good" if the evaluation standards are high, ensuring that the authors' ideas can circulate rapidly in the academic community. The implementation of exclusive criteria and standards (impact factor and article influence score, with minimum thresholds) for evaluating the research urges Romanian researchers to increase their scientific output, even if this goal is frequently achieved through speculative topics and meaningless statistics [27]. For Romania, this means traveling back in time some 200 years (1828), when one of the founders of Romanian Academy (Ion Heliade-Rădulescu), referring to the need to write in Romanian (replacing Slavic writing), said "Write, boys, just write". Indirectly, this imperative seems to be very topical (if rephrased as "Publish, boys, just publish"), because other economic and social impacts of research are totally neglected when evaluating the research and researchers.

### 4.7. Methodological Novelty and Advantages

The rank-based analysis used in this study yielded new findings, useful for a better understanding of the gaps in research performance between the Central and Eastern European countries, judged by rank dynamics. The adjusted citation index used as a relative measure of performance, taking into consideration the changes within a group of countries, with any assessment of the research value allowed for determining the impact of the publications of a country on the global academic community. Surprisingly, small countries (e.g., Macedonia, Azerbaijan, Armenia or Belarus), had in some cases the highest values of the index. This could be explained by the interest of the academic world in knowing more about "exotic" scientific areas. In this case, a few articles bring more citations.

### 4.8. Methodological Limitations

The analysis of individual research areas revealed strange patterns, in many cases due to the "small figures" effect. While the effect could be countered by the aggregation of domains, it allowed for pinpointing several shortcomings of the "citations per document" index. First of all, it does not reflect a precise timely situation, as it relates the number of articles published in a given year to the citation of articles from the previous years. Since the citation lifetime varies across different domains, the index can give only a general idea of the visibility. Second, if a country did not publish any articles, but received citations of the previous ones, the citation data are lost, as the division by zero prevents the computation of the number of citations per document. Unfortunately, the current study was not able to further elaborate on the analysis, making adjustments for the citation lifetime, given the lack of data on the citation lifetime by research area—especially with respect to SCImago's classification of the research areas. Instead, we have chosen a long period (nine years), which should be sufficient to counter this effect. Further research can focus on domains where the citation lifetime is known, eliminating this effect.

The key concept of our approach is the *relative research performance*, assessing the territorial differences between the countries within the analyzed area. We realize that we do not have sufficient indicators to guarantee that the quantitative performance is completely revealed. The number of articles per active researcher could be a good indicator for the scientific output. However, we experienced two problems when using it: first, it is difficult to find exact data for the number of active researchers by field for each country included in the study and for the entire period; and second, the definition of "researchers" varies from one country to another. Some Eastern European countries have national research institutes, reminiscent of the organization of research during the communist period; the term "researchers" includes these people and, to a lesser extent, the researchers from academia. In the universities, very few people are classified as "researchers", but all the academics (teaching personnel) carry out research activities. In addition, retired professors and researchers are also publishing. Being aware of the limits of the proposed index in relation to its ability to reflect the research performance, we labeled it "relative". While in this study we used an unadjusted number of documents, future studies might focus on countries where data is available, or find a realistic adjustment.

## 5. Conclusions

The observed changes of individual or aggregated sciences show an important diversity between states belonging to the same space; nevertheless, all of them witnessed the effects of the same centralized research policies. The East-West cooperation was correlated, on the one hand, with the openness to the market economy of the former socialist countries, and on the other with their access to some facilities offered by the developed ones. In this regard, the geographical distance played an important role, meaning that the countries in direct contact with the Western ones (including the Baltic republics) were the first to receive different research facilities: fellowships, joint contracts, special scientific cooperation programs, etc. This explains the advance of this first group of countries in publishing in good journals, having some domestic journals indexed and cited in prestigious databases, and working together

in different research consortia before their integration in the EU. As a consequence, most of these countries are in the first half of the regional ranking.

Another group is represented by Romania and Bulgaria, both with similar research policies on research, and among the last members of the EU. The new trend of measuring the efficiency of research activity only based on publishing seems ineffective in bettering the regional position of these countries. Some progress exists in Romania with respect to the social sciences, but not enough to bypass the impact of ten years of close cooperation inside the EU, which explains the progress of the first group.

The third and last group of countries is composed by the former Soviet countries, which started only later to develop research policies, with few resources and a huge handicap represented by their common communication language—different from English, the main publishing language. The fact that these countries have allocated few resources to research and that their scholars do not have the possibility to attend international meetings or constitute scientific cooperation networks explains their low positions in the regional rankings.

The extrapolation of the rank dynamics of the three countries selected as case studies suggests some trends, not necessarily valid as general conclusions for Central and Eastern Europe. The analyses revealed great differences across the Eastern European countries in terms of scientific visibility: "top players" (Slovenia), middle players (Poland) and countries at the end of the top (Romania). With time, the "top players" developed steadier trends, especially for several research areas which seem to be the effect of a "smart specialization". At the same time, the period 2010–2018 brought important changes compared to the whole period covered by the data (1996–2018): "top players" seem to show a consistent decrease for some fields (Slovenia), the countries situated at the end of the "top" seem to perform better in some areas (Romania) and the average players show inconsistent patterns, with some fields doing better and others worse (Poland). The analysis revealed the effect of a long research tradition in some fields, which perform better against the overall situation of the country. The three countries belonging to the EU benefited from different favorable tools and research cooperation frameworks, in comparison with the non-EU ones, but the slow pace of development and different R&D investment volume and research cooperation experiences explain their current positions in the regional ranking.

Nobody can evaluate the impact of the current trends only by using the number of papers published in the journals from the first two quartiles, the number of citations per article, the Hirsch index or other scientometric indexes as a measure of research performance. The negative effects of co-authorship networks will most likely reduce the possibility of isolated or young authors (especially from countries similar to the former socialist ones), with bright ideas, to publish their research due to less resources and notoriety. On the other hand, the less developed countries have specific problems, which should be addressed by new solutions, such as asking scholars for more applied research. The exaggeration of developing an evaluation system based exclusively on publishing could have unexpected impacts, blocking the creative solutions needed for solving local, regional and global problems.

In a nutshell, the take-home message of this study is that research performance can be achieved only by building up international research networks on solid grounds, rather than simply establishing citation networks; joint research programs and funding are an essential condition for their development. The importance of the international cooperation between the Western developed countries and the Central and European ones, with respect to the quality of research outputs in the latter, was proved by different scientometric studies [37,55]. At the same time, the Romanian example shows that, although the scientific output can be increased by restrictive policies—validating some of the results obtained by Fiala in his study on the effects of changing the national methodologies for institutional evaluation [33]—this does not automatically imply that the quality of research will increase. Sometimes such policies lead to unethical behaviors, such as the proliferation of self-citations and citation stacking.

Our study complements the information brought by similar articles and offers scholars a real image about the changes in large areas using only a rank-based analysis. We hope that the further development of this study will create the basis to bridge the current gaps between the countries, by intensifying the cooperation in scientific research.

**Author Contributions:** All the authors have equally contributed to the article. Conceptualization, I.I. and A.-I.P.; methodology, I.I. and A.-I.P.; formal analysis, A.-I.P. and I.I.; investigation, I.I. and A.-I.P.; resources, A.-I.P. and I.I.; data curation, I.I. and A.-I.P.; writing—original draft preparation, I.I. and A.-I.P.; writing—review and editing, A.-I.P. and I.I.; validation, I.I. and A.-I.P.; visualization, A.-I.P. and I.I.; supervision, I.I. and A.-I.P.; project administration, I.I. and A.-I.P. All authors have read and agreed to the published version of the manuscript.

**Funding:** This research received no external funding.

**Conflicts of Interest:** The authors declare no conflicts of interest.

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
