# Peer review of "An Overview of the Dynamics of Relative Research Performance in Central-Eastern Europe Using a Ranking-Based Analysis Derived from SCImago Data"

_publications, doi:10.3390/publications8030036_

Round 1

Reviewer 1 Report

This article entitled  “An overview on the dynamics of relative research performance in Central-Eastern Europe using ranking-based analysis derived from SCImago data” focuses on the analysis of research papers and citations of former communist bloc countries based on Scopus data from 2010-2018 and using a new bibliometric indicator that takes into account self-citations.

The topic is interesting and deserves investigation because the analysis of the research performance of Central and Eastern European countries is still underestimated. There is no doubt that the authors thoroughly explored the available data and spent a good deal of time analyzing and interpreting them. Even though there is no general conclusion because the results do not show a straightforward pattern (which the authors themselves admit), the study is of interest to the readers of Publications. However, there are several grave issues that must be resolved first. In the following, I list them in a decreasing order of importance:

Equations are not legible in the PDF file I obtained for review. In their place there is an error message (“Error! Objects cannot be created from editing field codes.”) on p. 3 and 4. For this reason I wasn’t able to check the main bibliometric indicator (adjusted number of citations) used in the study.

The charts are not especially enjoyable to look at. In my experience, a reasonable chart should not include more than five or six curves (data series) and, therefore, those charts with more than 20 of them (Figs. 2, 3, and 5) are practically useless even when printed in colour. Moreover, some data series appear to use the same line style and colour in print (for example like the Czech Republic and Slovakia in Figs. 2 and 3), which is really annoying. The authors should devise a smarter way of visualizing the results instead of putting it all into one line chart. What about splitting the set of data series into a couple of subsets or use a completely different type of charts? The authors should be more inventive here.

There is one additional chart for Slovenia (Fig. 5 on p. 10) that does appear neither for Poland nor for Romania. Why is that? Either add the same chart for both Poland and Romania too or remove it from Slovenia. It doesn’t make sense in the current form.

Even though there are a number of cited references in the paper, I missed two highly relevant ones on the topic of research outputs in the former Eastern Bloc countries:
doi:10.3390/soc3030266 and doi:10.1108/AJIM-02-2015-0027.

I don’t think that the term “auto-citation” (used in many places in the text) is an established bibliometric expression although it can be used marginally in some countries. Please use the term self-citation instead. Otherwise, the number of grammatical or spelling errors is low, but  the paper may be worth  of re-reading and proofreading in any case.

At some places in the paper the text looks too descriptive. (It is basically about plotting some charts and describing in detail what one can see in them.) Please try to lessen the descriptive part and amend the analytical (generalizing) part of the text. This would bring some genuine added value to the article.

Author Response

Thank you for your comments. The way of addressing them is detailed below:

R1
First of all, we would like to express our sincere gratitude to Reviewer #1 for the time
dedicated to the review and the comprehensive, deep and constructive remarks, which
allowed us to improve the quality of our article, deepening the research and potentially
helping it reach a broader audience. In this regard, we approached every comment, hopefully,
in a proper and exhaustive manner. The text below presents in detail the way of addressing
each comment; the references are to the final line numbers of the revised article. In addition,
the text was marked using a blue font, similar to the one used in this document to indicate
how each comment was addressed, in order to allow for easily identifying the changes.
This article entitled “An overview on the dynamics of relative research performance in
Central-Eastern Europe using ranking-based analysis derived from SCImago data” focuses
on the analysis of research papers and citations of former communist bloc countries based on
Scopus data from 2010-2018 and using a new bibliometric indicator that takes into account
self-citations.
The topic is interesting and deserves investigation because the analysis of the research
performance of Central and Eastern European countries is still underestimated. There is no
doubt that the authors thoroughly explored the available data and spent a good deal of time
analyzing and interpreting them. Even though there is no general conclusion because the
results do not show a straightforward pattern (which the authors themselves admit), the study
is of interest to the readers of Publications. However, there are several grave issues that must
be resolved first. In the following, I list them in a decreasing order of importance:
Equations are not legible in the PDF file I obtained for review. In their place there is an error
message (“Error! Objects cannot be created from editing field codes.”) on p. 3 and 4. For this
reason I wasn’t able to check the main bibliometric indicator (adjusted number of citations)
used in the study.
Thank you for this comment and apologies. The article was written on different computers,
using different Office versions, and the equations were lost due to the incompatibilities of
these versions. We will make sure that the equations are visible in the revised version; they
are inserted in lines 111 and 171.
The charts are not especially enjoyable to look at. In my experience, a reasonable chart should
not include more than five or six curves (data series) and, therefore, those charts with more
than 20 of them (Figs. 2, 3, and 5) are practically useless even when printed in colour.
Moreover, some data series appear to use the same line style and colour in print (for example
like the Czech Republic and Slovakia in Figs. 2 and 3), which is really annoying. The authors
should devise a smarter way of visualizing the results instead of putting it all into one line
chart. What about splitting the set of data series into a couple of subsets or use a completely
different type of charts? The authors should be more inventive here.
Thank you for your suggestion. We totally agree, and had trouble in the first place; with the
help of statistics, we opted out for an aggregated representation, using boxplots for each series
instead of plotting out all its values.
There is one additional chart for Slovenia (Fig. 5 on p. 10) that does appear neither for Poland
nor for Romania. Why is that? Either add the same chart for both Poland and Romania too or
remove it from Slovenia. It doesn’t make sense in the current form.
Thank you for your suggestion. Since the comments on the results existed already, we decided
to add similar charts for Poland and Romania.
Even though there are a number of cited references in the paper, I missed two highly relevant
ones on the topic of research outputs in the former Eastern Bloc countries:
doi:10.3390/soc3030266 and doi:10.1108/AJIM-02-2015-0027.
Thank you! It’s nice to have such recommendations. Your suggestions improve our literature
review and helped validating some of our hypotheses. We have cited both papers in two
places each one, because the articles really support our conclusions.
I don’t think that the term “auto-citation” (used in many places in the text) is an established
bibliometric expression although it can be used marginally in some countries. Please use the
term self-citation instead. Otherwise, the number of grammatical or spelling errors is low, but
the paper may be worth of re-reading and proofreading in any case.
The term was replaced as suggested. In addition, the article was proofread in its revised form.
At some places in the paper the text looks too descriptive. (It is basically about plotting some
charts and describing in detail what one can see in them.) Please try to lessen the descriptive
part and amend the analytical (generalizing) part of the text. This would bring some genuine
added value to the article.
Thank you for this suggestion. However, since the structures of articles, imposed by
“Publications”, requires a separation of results and discussions, we cannot develop too much
the result sections, confined to presenting the results and pointing to the reader the most
important findings in each table or figure. Therefore, in order to address the comment, we
have significantly developed the discussions, adding new sections (4.2, 4.5) and elaborating
on the existing ones. Moreover, if the older presentation of results was meant to say in words
what was not visible (i.e., in Microsoft Excel we were able to click and follow particular data,
which were hard to discern in the older charts), now such trends are no longer visible, and the
descriptive part complements better the new graphs.

Reviewer 2 Report

The authors address the development of a Central-Eastern European centered metric to reflect the cultural and scholarly relationships that exist between the countries and provide additional context to other studies.

The article provided an interesting discussion of an adjusted citation rate. Please consider the following for improvement

p2:48: "pseudo-research elites"- if not opinion, this needs citation

p2: 60: "The recent literature includes a lot of articles" - recommend a  number for comparative purposes (10, hundreds, more than 500?)

p2:77 - Note in the introduction "scholars usually take into consideration two databases, i.e. those offered by 56 Clarivate Analytics and Scopus,... " why did the authors choose to only look at Scopus?

p.3 equation missing, please clarify where auto-citation rate. From the discussion it is unclear how it is different from the citations.

Figure1: The line graph is not the best format to visualise this information.

3.3: As written, auto-citation and self-citation are used interchangeably. This may tie to p.3 comments. please define auto-citation (having spent time examining auto-citations as tools that create citations or link references, self-citations seems to fit model - could have been clarified in equation that is not visible) 

Figures in section 3. the authors note that figure 2 is chaotic. The figures do not add to the discussion, too many variables are included in each table.

3.5.3 the dictator’s wife needs citations (interesting anecdote)

4.4. Synthetic - should this be Synthesis? Please review English. There are areas where the authors talk around a concept without defining (auto-citation would be an example). Another example: "occupants of the top positions"  = highly ranked countries?

Is the 2010-2018 window enough of a time frame for discussion? 

Authors note in limitations: "Since the citation lifetime varies across different domains, the index can give only a general idea of the visibility" however, the discussions in the cases note the low citations rates without noting differences in lifetime. Consider grouping disciplines around citation lifetime for comparisons. Or just focus on a specific lifetime that makes sense within the time identified. Comparisons across cases would be helpful as well - rather than comparing engineering and humanities for Slovenia, the authors may strengthen their case by comparing disciplines between the cases.

While the authors noted the change over time. What is the difference between the citation rate and the adjusted citation rate? Would this be the difference in Table 3? Perhaps sorting on one of the variables rather than presenting in alphabetical order. The authors could provide more discussion or clarify this table. 

Author Response

Thank you for your comments. The way of addressing them is detailed below

R2
First of all, we would like to express our sincere gratitude to Reviewer #2 for the time
dedicated to the review and the comprehensive, deep and constructive remarks, which
allowed us to improve the quality of our article, deepening the research and potentially
helping it reach a broader audience. In this regard, we approached every comment, hopefully,
in a proper and exhaustive manner. The text below presents in detail the way of addressing
each comment; the references are to the final line numbers of the revised article. In addition,
the text was marked using a blue font, similar to the one used in this document to indicate
how each comment was addressed, in order to allow for easily identifying the changes.
The authors address the development of a Central-Eastern European centered metric to reflect
the cultural and scholarly relationships that exist between the countries and provide additional
context to other studies.
The article provided an interesting discussion of an adjusted citation rate. Please consider the
following for improvement
p2:48: "pseudo-research elites"- if not opinion, this needs citation
This is an opinion, but the idea was also published in one of the articles already cited; we
inserted the reference to it. In addition, we have elaborated on the idea a little bit more,
rephrasing and developing the entire paragraph (lines 48-52).
p2: 60: "The recent literature includes a lot of articles" - recommend a number for
comparative purposes (10, hundreds, more than 500?)
Quantified as suggested: “few dozens”. We have already cited the most relevant at the end of
phrase (line 61).
p2:77 - Note in the introduction "scholars usually take into consideration two databases, i.e.
those offered by 56 Clarivate Analytics and Scopus,... " why did the authors choose to only
look at Scopus?
Thank you for this comment. We have motivated our choice in the first paragraph of the
methodology showing that only Scopus has aggregated data for institutions and countries
(lines 85-86).
p.3 equation missing, please clarify where auto-citation rate. From the discussion it is unclear
how it is different from the citations.
Thank you for this comment and apologies. The article was written on different computers,
using different Office versions, and the equations were lost due to the incompatibilities of
these versions. We will make sure that the equations are visible in the revised version; they
are inserted in lines 111 and 171.
Figure1: The line graph is not the best format to visualise this information.
We hope that presenting the graph as a bar chart would accommodate the Reviewer’s request.
3.3: As written, auto-citation and self-citation are used interchangeably. This may tie to p.3
comments. please define auto-citation (having spent time examining auto-citations as tools
that create citations or link references, self-citations seems to fit model - could have been
clarified in equation that is not visible)
As requested by another Reviewer, we used the term “self-citation” everywhere in the article.
Additionally, we provided the SCImago definition of self-citations (lines 107-109).
Figures in section 3. the authors note that figure 2 is chaotic. The figures do not add to the
discussion, too many variables are included in each table.
Thank you for your suggestion. We totally agree, and had trouble in the first place; with the
help of statistics, we opted out for an aggregated representation, using boxplots for each series
instead of plotting out all its values.
3.5.3 the dictator’s wife needs citations (interesting anecdote)
Thank you for your suggestions. We elaborated a little bit on the story (since the source was
in Romanian) in lines 440-444 and provided a citation of the most relevant article about how
she obtained her PhD degree (reference no. 41).
4.4. Synthetic - should this be Synthesis? Please review English. There are areas where the
authors talk around a concept without defining (auto-citation would be an example). Another
example: "occupants of the top positions" = highly ranked countries?
Thank you for your suggestions. We replaced “synthetic” by “brief” (line 56*, and “occupants
of the top positions” by “highly ranked countries” (line 299).
Is the 2010-2018 window enough of a time frame for discussion?
We believe than nine years are sufficient for compensating all effects described in the
concerns of the Reviewer. We have also discussed it in relationship to the citation lifetime in
the first paragraph of section 4.8 (lines 642-647).
Authors note in limitations: "Since the citation lifetime varies across different domains, the
index can give only a general idea of the visibility" however, the discussions in the cases note
the low citations rates without noting differences in lifetime. Consider grouping disciplines
around citation lifetime for comparisons. Or just focus on a specific lifetime that makes sense
within the time identified. Comparisons across cases would be helpful as well - rather than
comparing engineering and humanities for Slovenia, the authors may strengthen their case by
comparing disciplines between the cases.
The comment is more than welcome, but unfortunately almost impossible to address,
provided the lack of data on the citation lifetime by field – especially with respect to
SCImago’s classification of field. We have elaborated on this issue in the discussions (lines
642-647). As for the comparisons of disciplines, section 3.4.1 deals with exactly this type of
analysis, looking at two research areas (dentistry, among the individual ones, and engineering
sciences, among the aggregated ones).
While the authors noted the change over time. What is the difference between the citation rate
and the adjusted citation rate? Would this be the difference in Table 3? Perhaps sorting on one
of the variables rather than presenting in alphabetical order. The authors could provide more
discussion or clarify this table.
New Table 3 includes results addressing two issues discussed in the revised article (the
scientific output and its visibility). For this reason, sorting for one variable would lean the
balance for one of the two, while our intention is to address them equally. However, we have
clarified the column labels, corrected the explanation of the table, and improved its discussion
in section 3.3 (lines 240-244).

Reviewer 3 Report

The manuscript by Ianos and Petrisor describes an analysis of the regional research performance dynamics of central-eastern European countries based on adjusted numbers of citations per document and related these findings to historical facts and current developments affecting the research environment in individual countries.

Comments/suggestions:

  • the equations (1) and (2) are not included in the reviewed version of the manuscript.
  • the layout of table 1 could be optimized to clearly indicate which variables are correlated with each other.
  • The “small figures” effect has been correctly mentioned in the limitations section – against this background, the statement (pg. 11, line 368) on an “ascending trend” might not be valid.
  • The central variable used in this study is numbers of citations (per no. documents) roughly indicating visibility of the published research and rank positions based on this figure. Due to factors generally affecting the selection of citations (exemplified for auto-citations in section 4.2.), the chosen approach might be valid for judging on the visibility of published research but not necessarily for general performance (as suggested on pg. 14, line 504). Therefore, a second line of analyses might be included in the manuscript which is based on the on the research output (number of articles/documents per year/per country/per field), probably adjusted for the number of active researchers in the respective scientific field. This would exclude distorting effects inherent to citation counts (perception of scientific content) and provide a more direct and interpretable picture of scientific performance.
  • For demonstrating the temporal dynamics, i.e. changes in rank position, the current design of figures (no. 2-7) is not optimal: individual countries are hardly distinguishable (and impossible to read on a black-white printout). Additionally, the authors might consider including a number quantifying the variability of the (changing) rank, for example the coefficient of variation – these numbers could be shown in a separate diagram indicating the rank variability per research field and/or country.
  • Although I am not an expert regarding the effects of (historical) national research policies on research performance (indicators), I find it difficult to derive general statements (as in abstract, line 22/23, and discussion, line 569-575) on a country level for all disciplines of research. Instead, field-, country- and period-specific models of explanation (as exemplary provided on pg. 10/11, line 352-357) might be more appropriate.

Author Response

Thank you for your comments. The way of addressing them is detailed below:

R3
First of all, we would like to express our sincere gratitude to Reviewer #3 for the time dedicated to
the review and the comprehensive, deep and constructive remarks, which allowed us to improve the
quality of our article, deepening the research and potentially helping it reach a broader audience. In
this regard, we approached every comment, hopefully, in a proper and exhaustive manner. The text
below presents in detail the way of addressing each comment; the references are to the final line
numbers of the revised article. In addition, the text was marked using a blue font, similar to the one
used in this document to indicate how each comment was addressed, in order to allow for easily
identifying the changes.
The manuscript by Ianos and Petrisor describes an analysis of the regional research performance
dynamics of central-eastern European countries based on adjusted numbers of citations per
document and related these findings to historical facts and current developments affecting the
research environment in individual countries.
Comments/suggestions:
· the equations (1) and (2) are not included in the reviewed version of the manuscript.
Thank you for this comment and apologies. The article was written on different computers, using
different Office versions, and the equations were lost due to the incompatibilities of these versions.
We will make sure that the equations are visible in the revised version; they are inserted in lines 111
and 171.
· the layout of table 1 could be optimized to clearly indicate which variables are correlated
with each other.
The layout was optimized as suggested, including on both horizontal and vertical axes all
variables and indicating the results of the correlation analysis at their intersection.
· The “small figures” effect has been correctly mentioned in the limitations section – against
this background, the statement (pg. 11, line 368) on an “ascending trend” might not be valid.
Thanking for your remark, we reconsider our initial conclusion (line 462). This is not just a
trend, but rather an accident, especially taking into consideration that it’s about only one year.
We have modified the phrase.
· The central variable used in this study is numbers of citations (per no. documents) roughly
indicating visibility of the published research and rank positions based on this figure. Due to
factors generally affecting the selection of citations (exemplified for auto-citations in section
4.2.), the chosen approach might be valid for judging on the visibility of published research
but not necessarily for general performance (as suggested on pg. 14, line 504). Therefore, a
second line of analyses might be included in the manuscript which is based on the on the
research output (number of articles/documents per year/per country/per field), probably
adjusted for the number of active researchers in the respective scientific field. This would
exclude distorting effects inherent to citation counts (perception of scientific content) and
provide a more direct and interpretable picture of scientific performance.
The suggestion is very good, but unfortunately data on the number of researchers per field,
country, and year are not available for all the countries studied. As a matter of fact, we had to be
quite innovative even for a simpler analysis. The methodological section describes this issue in
relationship to the influence of a critical mass of researchers and research expenditure. Only 16
countries have data available on Eurostat, with one of them (the Russian Federation) only for
part of the study period. Using data from other sources would certainly raise additional concerns
in validity terms.
While we have stressed out these limitations, we carried out an analysis of the scientific output
based on the number of documents, and we are really indebted to Reviewer 3 for this
suggestion, because it revealed very interesting insights. Puzzled by the differences between the
rankings based on the scientific output and those based on the visibility, we decided to run
additional analyses, described in new sections (3.5 for the results, and 4.2 and 4.5 for the
discussion); we believe that, although overall inconclusive, these results may constitute the start
point for further research, and reveal the many differences across countries and research
domains.
· For demonstrating the temporal dynamics, i.e. changes in rank position, the current design
of figures (no. 2-7) is not optimal: individual countries are hardly distinguishable (and
impossible to read on a black-white printout). Additionally, the authors might consider
including a number quantifying the variability of the (changing) rank, for example the
coefficient of variation – these numbers could be shown in a separate diagram indicating the
rank variability per research field and/or country.
Thank you for your suggestion. We totally agree, and had trouble in the first place; with the help
of statistics, we opted out for an aggregated representation, using boxplots for each series
instead of plotting out all its values. Boxplots show the variability and are easier to understand
by people without a quantitative background than a table of individual statistics, such as the
coefficient of variation.
· Although I am not an expert regarding the effects of (historical) national research policies on
research performance (indicators), I find it difficult to derive general statements (as in
abstract, line 22/23, and discussion, line 569-575) on a country level for all disciplines of
research. Instead, field-, country- and period-specific models of explanation (as exemplary
provided on pg. 10/11, line 352-357) might be more appropriate.
We appreciate this very good comment! Our discussion is about the adaptive versus very
restrictive policies; the latest determine an increase of the number of articles, self-citations and
pseudo-citations (through citation stacking networks), with a really poor impact. The phrase was
radically changed (lines 596-600).

Round 2

Reviewer 1 Report

I am quite satisfied with the amendments to the article made by the authors and thank them for their effort to improve the paper. All issues raised by me were carefully addressed in the revised version and, to the extent I can judge, also all the concerns of the other two reviewers.

I have only noticed the following minor errors:

Please remove the spaces from "self - citations" in Eq. (1) on line 111 because it looks confusing.

Please replace "the number of citation" with "the number of citations" on line 504.

Please replace "to explain while Romania is situated" with "to explain why Romania is situated" on line 597.

Please replace "very few people are classified by" with "very few people are classified as" on line 657.

I'm still missing axis labels from Fig. 2 onward. (Interestingly, they are present in Fig. 1 only.)